# Accumulation of Paneth Cells in Early Colorectal Adenomas Is Associated with Beta-Catenin Signaling and Poor Patient Prognosis

**DOI:** 10.3390/cells10112928

**Published:** 2021-10-28

**Authors:** Erika López-Arribillaga, Bing Yan, Teresa Lobo-Jarne, Yolanda Guillén, Silvia Menéndez, Montserrat Andreu, Anna Bigas, Mar Iglesias, Lluís Espinosa

**Affiliations:** 1Stem Cells and Cancer Research Laboratory, Institut Hospital del Mar d’Investigacions Mèdiques (IMIM), CIBERONC, 08003 Barcelona, Spain; Erika.lopez@irbbarcelona.org (E.L.-A.); byan@imim.es (B.Y.); tlobo@imim.es (T.L.-J.); yguillen@imim.es (Y.G.); abigas@imim.es (A.B.); 2Departament of Oncology, Hospital del Mar, 08003 Barcelona, Spain; smenendez@imim.es; 3Gastroenterology Department, Hospital del Mar, IMIM (Hospital del Mar Medical Research Institute), 08003 Barcelona, Spain; montserrat.Andreu@psmar.cat; 4Departament of Pathology, Hospital del Mar, CIBERONC, 08003 Barcelona, Spain

**Keywords:** colorectal cancer, adenomas, stem cell niche, Paneth cells, Wnt, Notch

## Abstract

Background: Previous studies in mice indicated that Paneth cells and c-Kit-positive goblet cells represent the stem cell niche of the small intestine and colon, respectively, partly by supporting Wnt and Notch activation. Whether these cell populations play a similar role in human intestinal cancer remains unexplored. Methods: We performed histopathological evaluation and immunohistochemical analysis of early colorectal adenomas and carcinoma adenoma from patients at the Hospital del Mar in Barcelona. We then determined the possible correlation between the different parameters analyzed and with patient outcomes. Results: Paneth cells accumulate in a subset of human colorectal adenomas directly associated with Notch and Wnt/β-catenin activation. Adenoma areas containing Paneth cells display increased vessel density in the lamina propria and higher levels of the stem cell marker EphB2. In an in-house cohort of 200 colorectal adenoma samples, we also observed a significant correlation between the presence of Paneth cells and Wnt activation. Kaplan–Meier analysis indicated that early adenoma patients carrying Paneth cell-positive tumors display reduced disease-free survival compared with patients with Paneth cell-free lesions. Conclusions: Our results indicate that Paneth cells contribute to the initial steps of cancer progression by providing the stem cell niche to adenoma cells, which could be therapeutically exploited.

## 1. Introduction

Colorectal cancer (CRC) is the second leading cause of death by cancer in the industrialized countries. However, most CRC patients do not die because of the primary lesions that are generally removed by surgery, but as a consequence of more aggressive invasive or metastatic tumors. Tumor relapse after therapy and tumor metastasis mostly depend on an exclusive population of cells known as “tumor initiating cells” or cancer stem cells (CSCs) that share specific features with normal intestinal stem cells (ISC) including their transcriptional profiles [1].

In the normal mouse intestine, a limited number or Paneth cells constitute the niche of the ISC populations residing at the bottom of the crypts [2], which are characterized by high EphB2 levels and the combined expression of the stem cell markers Bmi1, Lrig, mTert, Hopx or Lgr5 [3,4,5,6,7]. Paneth cells are responsible for providing ISC with the signals required to activate the Notch and Wnt/β-catenin pathways [2] that are essential for stem cell homeostasis. However, in mouse models displaying Paneth cell depletion, enteroendocrine and tuft cells can serve as an alternative source for these signals, thus supporting ISC maintenance [8]. In the mouse colon, there is a rare population of Goblet cells that express the stem cell factor receptor c-Kit and exert this ISC niche function in the absence of Paneth cells [9]. Moreover, c-Kit expression upon inflammation triggers signaling through the c-Kit receptor leading to Paneth cells re-entering into cell cycle, thus contributing to the regeneration process [10]. Interestingly, Paneth cell reprogramming by damage is dependent on Notch signaling [11].

Multiple evidence indicates that tumor-initiating cells depend on the same signaling pathways that regulate normal ISC function including Notch [12,13,14,15,16,17,18], Wnt//β-catenin [19,20] and TGF-β [21], through activation of a stem cell transcriptional program containing genes such as *Lgr5*, *Bmi1*, *EphB2* or *Hes1*.

Using a mouse model of intestinal cancer that carries mutations in the APC gene (the principal negative regulator of Wnt/β-catenin), we previously demonstrated that aberrant activation of β-catenin in the adenoma cells induced transcription of the Notch ligand Jagged1 leading to Notch signaling [22], which was imposed by defective Manic Fringe function [18]. A similar mechanism linking Notch signaling with β-catenin function was found to operate in human tumors from Familial Adenomatous Polyposis patients [22].

Apart from being the source of stem cell signals in the small intestine, Paneth cells have previously been associated with β-catenin pathway activation in human adenomas [23] and a specific Paneth cell population expressing indoleamine-2,3-dioxygenase-1 was found to promote immune evasion in human colorectal cancer [24]. However, it remains controversial whether and how the presence of Paneth cells positively or negatively impact CRC malignancy [24,25,26,27,28].

Here, we investigate the relationship between Notch and β-catenin signaling and the presence of Paneth cells in human colorectal tumors, and its possible contribution to tumor malignancy.

## 2. Materials and Methods

### 2.1. Patients

Samples of 38 patients from the Hospital del Mar (Barcelona, Spain) with pathologically determined colorectal adenocarcinomas were initially selected for this study based on the presence or absence of Paneth cells. General information about these tumors is shown in Appendix A. The study protocol was approved by the Ethics Committee of Human Resources at Hospital del Mar, written informed consent was obtained from the patients. We obtained an additional set of 200 formalin-fixed, paraffin-embedded and randomly-selected tissue blocks of colorectal tumors from the archives of the Bank of Tumors of the Hospital del Mar (MarBiobanc, Barcelona, Spain). Multiple areas of invasive carcinoma, adenomatous polyps, and normal adjacent mucosa from the same surgical sample were identified on the corresponding hematoxylin and eosin-stained slides. The selected tissues were transferred to a recipient “master” block using a Tissue Microarrayer. Each core was 0.6-mm wide spaced 0.7–0.8 mm apart.

### 2.2. Immunohistochemistry (IHC)

In brief, the tissues were fixed in 4% paraformaldehyde, embedded in paraffin, and processed by standard histological methods. From each selected paraffin block, 4 μm serial sections were cut. Immunohistochemical (IHC) studies were performed with avidin-biotin-peroxidase complex kits according to the manufacturer’s instructions (Agilent, Dako, CA, USA). Primary antibodies (anti-EphB2 [rabbit polyclonal, dilution 1:200, R&D Systems, MN, USA AF467], anti-β-catenin [rabbit polyclonal, dilution 1:250, Sigma-Aldrich, Merk, Germany c2206], anti-LYZ [rabbit polyclonal, dilution 1:5000, Agilent, Dako, CA, USA. A0099], anti-c-kit [rabbit polyclonal, dilution 1:200, Santa Cruz Biotechnology, Heidelberg, Germany sc-5535], anti-synaptophysin [rabbit polyclonal, dilution 1:200, Agilent, Dako, CA, USA. A0010] and anti-CD31 [rabbit polyclonal, dilution 1:200]) were incubated in a humidified chamber at 4 °C overnight. After that, the samples were developed with liquid DAB + substrate chromogen system (Agilent, Dako, CA, USA) under microscope (DM3000, Leica, Wetzlar, Germany) and the results were counted by two independent researchers. The method of IHC READING calculation was as follows: Staining intensity: 0 = no staining, 1 = weak staining, 2 = moderate staining, 3 = strong staining; percentage of positive cells: 0 = 0%, 1 = 1–10%, 2 = 10–50%, 3 ≥ 50%.

### 2.3. Double Immunofluorescence

Briefly, after the samples were incubated with primary antibody (anti-β-catenin [rabbit polyclonal, dilution 1:250, Sigma-Aldrich, Merck, Darmstadt, Germany c2206], anti-LYZ [rabbit polyclonal, dilution 1:5000, Agilent, Dako, CA, USA A0099], anti-ICN1 [rabbit monoclonal, dilution 1:200, Cell Signaling Technology, MA, USA #4147], anti-Bmi1 [rabbit monoclonal, dilution 1:200, Cell Signaling #6964]), the samples were washed and then developed with Tyramide Signal Amplification (TSA™) plus the Cyanine 3/Fluorescence system (NEL753001KT, PerkinElmer, MA, USA) under the microscope (BX51, Olympus) and then mounted with DAPI (H-1200, Vector Laboratories, CA, USA); the positive results were stained green or red and the pictures were taken by confocal laser scanning microscopy (STP-6000, Leica, Wetzlar, Germany).

### 2.4. Analysis of the IHC Data

Two researchers without knowledge of the clinical data independently evaluated the IHC results. In all the analyses a minimum of five fields containing tumour areas were randomly selected and evaluated by two independent researchers. Images at 400× magnification were used to determine the percentage of tumor cells that exhibited positive immunoreactivity for the different antibodies. In tissue microarray samples, staining intensity was assessed directly under the microscope by two independent researchers, at 100× magnification, but amplifying tumour areas at 200× when necessary to obtain agreement among researchers. In the case of Notch and β-catenin, nuclear and cytoplasmatic stainings were evaluated separately. Intensity of positive staining was defined as follows: light: 1, moderate 2, intense 3. The value for negative staining was 0. The Histoscore index defined as (0×%) + (1×%) + (2×%) + (3×%) was used to analyze the correlations between IHC results and clinical information including the pathological grade and TNM stage.

Pearson’s chi-square test with a cross tabulation table of 2 × 2 was performed to determine the statistical association between marker expression (positive or negative) using Excel. R score for correlation (Pearson coefficient) was used to test the correlation between quantitative measurements of marker expression (0-1-2-3 or beta-catenin and 0-1 for lysozyme) using Graphpad Prism 9; *p* value was calculated using a two-tailed confidence interval of 95%. Survival analyses were performed on Graphpad Prism 9, to measure 5-year disease-free survival; X values were measured in months, Y values were coded 1 for death/event, 0 for censored subjects; survival curves were compared using the logrank (Mantel–Cox test) and the Gehan–Breslow–Wilcoxon test.

## 3. Results

### 3.1. A Subset of Colorectal Adenomas Accumulates Paneth Cells That Are c-Kit and Synaptophysin-Positive 

Paneth cells are mostly located at the bottom of the small intestinal crypts in both mouse and humans. They are primarily absent from the normal colonic mucosa but their frequency increases in several inflammatory diseases [29]. We have here selected a small discovery cohort of 38 early adenomas characterized by the presence of large eosinophilic granules in the cytoplasm, indicative of Paneth cells, in about half of the samples (Appendix A). Notably, eosinophilic granules were specifically detected in the pseudo-stratified crypts of the tumor area, but excluded from the monolayered adenomatous glands present in the same samples (Figure 1A). Paneth cell identity was confirmed by IHC of sequential paraffin sections using a specific monoclonal antibody against the Paneth cell marker lysozyme (LYZ) (Figure 1B).

Previous studies indicated that Paneth cells also express synaptophysin [30]. Consistent with this observation, tumors classified as Paneth cell-positive expressed high levels of synaptophysin when compared with Paneth-negative samples (Figure 1C and Appendix A). In addition, we detected a significant amount of tumor cells displaying membranous c-kit expression in the Paneth cell-positive tumors (Figure 1D), together with an increased number of CD31-positive blood vessels, likely associated with the inflammatory environment of tumors. However, we did not find any association between proliferation index as determined by ki67 staining and Paneth cell positivity (Appendix A).

Together these results indicate that Paneth cells accumulate in the pseudostratified adenomas whereas they are absent from adjacent monolayered glands, thus suggesting a role for Paneth cells in early tumor progression.

### 3.2. Wnt/β-Catenin Activity Is Increased in Paneth Cell-Enriched Adenoma Areas

Paneth cells represent the stem cell niche in the small intestine by providing the source for Wnt and Notch activation [2]. Here, we investigated whether the presence of Paneth cells was associated with Wnt activation in human adenoma lesions. By IHC analysis, we noticed a significant accumulation of nuclear β-catenin (indicative of Wnt activation) restricted to the tumor areas containing Paneth cells (Figure 2A). Moreover, double immunofluorescence of LYZ and β-catenin demonstrated a perfect match between presence of Paneth cells and nuclear accumulation of β-catenin in these samples. Of note that nuclear β-catenin was located in both the Paneth cell population and the adjacent tumor cells, with no evidences of β-catenin activation in crypts lacking Paneth Cells (even in the same tumor area) (Figure 2B). Unexpectedly, we detected nuclear β-catenin staining in one of the samples that were initially considered negative for the presence of Paneth cells. Double IHC for β-catenin and LYZ demonstrated that nuclear β-catenin was restricted to few Paneth cells present in this sample and cells localized adjacent to them.

We then extended this study using a tissue microarray (TMA) containing 250 human samples including normal tissue (54), adenoma (143) and carcinoma areas (53) from 60 different patients. We detected LYZ-positive cells (considering positive samples classified as ++ and +++) in about 40% of the adenoma samples, covering different areas of the tumor tissue. This proportion was highly reduced in the more advanced carcinoma samples and almost negligible in the normal mucosa (Figure 2C). As expected, nuclear β-catenin detection was higher in carcinomas than in adenomas (Figure 2C) but we detected a significant correlation between the presence of nuclear β-catenin and LYZ positivity in the adenoma samples (Figure 2D).

These results indicate that Wnt/β-catenin activation is associated with LYZ expression during early colorectal tumorigenesis.

### 3.3. Notch1 Activity Is Higher in Adenoma Cells of the Paneth Enriched Areas

Next, we determined the possible association between Notch1 activation and the presence of Paneth cells and LYZ expression in colorectal tumors. IHC analysis of samples using the ICN1 antibody demonstrated higher Notch1 activity in adenoma and carcinoma samples compared with normal adjacent mucosa of the same patients (Figure 3A). Remarkably, adenoma glands enriched in LYZ-positive Paneth cells displayed higher ICN1 levels (yellow arrowheads) when compared with the adjacent LYZ-negative glands of the same tumor area (blue arrowheads) (Figure 3B). Notably, LYZ-positive tumors displayed a robust ICN1 staining in both the Paneth cell population and the adjacent (non-Paneth) adenoma cells. We also detected variable levels of ICN1 staining in samples devoid of Paneth cells indicating that Notch activation occurs by both Paneth cell-dependent [31] and -independent [18,22] mechanisms.

### 3.4. Expression of EPHB2 Is Linked with Wnt/β-Catenin Activation in Paneth Cell-Positive Adenomas

Previous studies indicated that Notch and Wnt/β-catenin, and specific combinations of these signals, modulate the proliferative rate of the murine intestinal stem cell compartment [32,33], which is characterized by EphB2 expression. EphB2 is also a robust marker of intestinal tumor-initiating cells [1]. By IHC analysis, we detected EphB2 staining in both Paneth cell-positive and -negative samples (determined by detection of eosin-positive granules) with some accumulation in Paneth cell-negative tumor samples (Figure 4A). However, EphB2 staining was accumulated in pseudostratified intestinal glands (in both types of tumor) and in Paneth cell-positive samples containing high nuclear β-catenin, being monolayered glands EphB2 negative (Figure 4A,B). Statistical analysis of data demonstrated a significant correlation (*p* < 0.001) between EphB2 levels and nuclear β-catenin that was restricted to Paneth-positive samples (Figure 4B).

All these results suggest that Paneth cells constitute the niche of early adenoma cells. There is increasing evidence that targeting the stem cell niche alone or in addition to treating tumor cells might represent a promising therapeutic option. In favor of this concept, non-tumor cell populations are less likely to develop resistance to therapy when compared to cancer cells, which display higher proliferation rates and mutational activity.

### 3.5. Presence of Paneth Cells in the Adenomas Is Indicative of Poor Prognosis

The demonstration that Paneth cell accumulation specifically in the adenoma samples correlating with higher levels of nuclear β-catenin and ICN1, and differential EPHB2 distribution led us to study the possibility that Paneth cells may serve as a prognosis marker in low-grade tumor patients. Our results indicated that patients carrying adenomas that were positive for Paneth cells (determined by histological detection of eosin-positive granules) showed reduced overall survival (*p* = 0.22; HR = 2.0) compared with those classified as Paneth-negative (Figure 5A). Similarly, patient stratification based on LYZ mRNA levels using the stage II Kemper CRC dataset [34] denoted a poor prognosis value of this marker (*p* = 0.30; HR = 3.35) (Figure 5B) that was not observed in the TCGA CRC dataset (Figure 5C). Interestingly, restricting the analysis to stage II patients in the TCGA dataset patients, LYZ-high tumors show a robust trend towards poor prognosis in the first 40 months after diagnosis (Figure 5D). Analysis of overall survival in this initial period confirmed the prognosis value of the LYZ marker (Figure 5E).

Together our data indicate the Paneth cells may contribute to the activation of essential stem cell signaling pathways at early colorectal adenomas, and suggest the usefulness of using Paneth cell detection for patient prognosis at early stages of tumor progression. However, further studies should be performed, including a higher number of patients, to obtain robust conclusion on the validity of Paneth cell detection in early adenomas as a criterion to refine patient guidance.

## 4. Conclusions

There are controversial data on the prognosis value of Paneth cell detection in colorectal tumors. Our results indicate that Paneth cell detection and LYZ expression is correlated with Wnt/β-catenin and Notch activation specifically in low-grade adenoma samples. Most importantly, the detection of Paneth cells in this group of tumors is directly associated with poor patient outcomes. The fact that Paneth cell differentiation is significantly reduced in the more advanced carcinoma samples has certainly made difficult the identification of Paneth cells as a poor prognosis factor in the less-advanced lesions. However, we have demonstrated that restricting the analysis to initial tumor stages (i.e., stage II) is sufficient to confirm the prognosis value of LYZ levels in CRC.

We propose that evaluating the presence/absence of mature Paneth cells in the adenoma samples at the time of diagnosis by detection of eosin-positive granules in the H&E-stained samples will provide the clinicians with additional information, which could directly impact patient prognosis with subsequent therapeutic implications.

## Figures and Tables

**Figure 1 cells-10-02928-f001:**
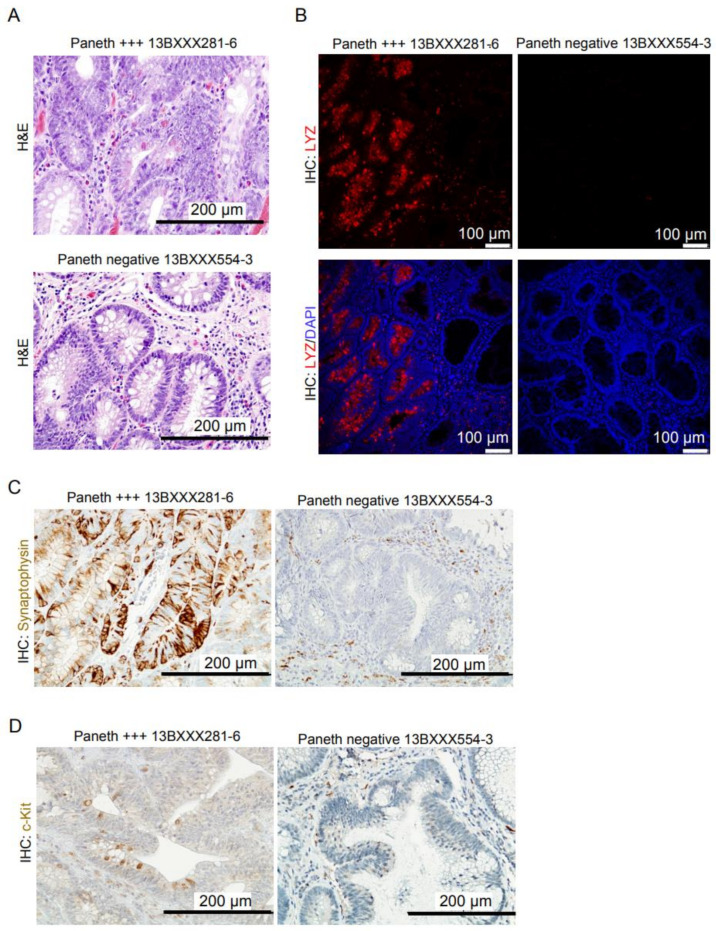
Adenomas accumulates Paneth cells that are c-Kit and synaptophysin-positive. (**A**) Representative hematoxylin and eosin staining of adenoma samples considered as Paneth cell-positive and -negative. (**B**) Immunofluorescence (IF) analysis of Paneth cell-positive and -negative samples with the Paneth cell marker lysozyme (LYZ). DAPI staining was used for nuclei detection. (**C**,**D**) Immunohistochemical (IHC) analysis of synaptophysin (**C**) and c-Kit markers (**D**) in the indicated samples. +++ indicates the degree of positivity in this particular samples as determined by the pathologists.

**Figure 2 cells-10-02928-f002:**
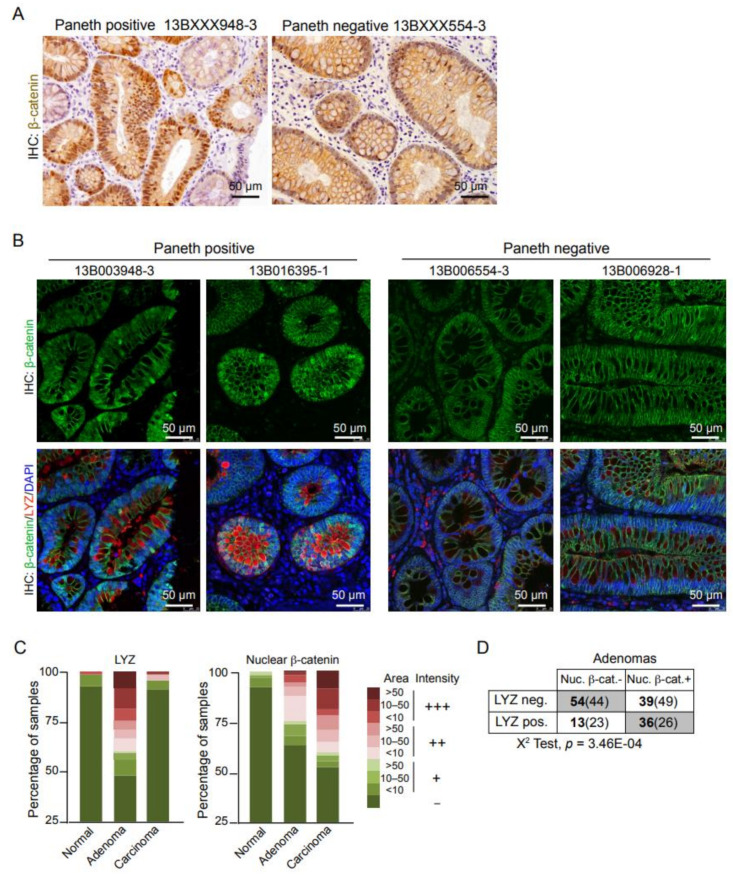
Wnt/β-catenin activity is increased in Paneth cell-enriched adenoma areas. (**A**) Immunohistochemical (IHC) analysis with anti-β-catenin antibody. Representative images of Paneth cell-positive and -negative samples are shown. (**B**) Double immunofluorescence (IF) analysis of β-catenin and LYZ in the indicated samples. DAPI was use for nuclei staining. (**C**) Quantification of the intensity of staining and area of the tissue (normal of tumoral) that was considered positive for the indicated markers. (**D**) Correlation analysis of the presence of nuclear β-catenin and LYZ positivity in the adenoma samples. The actual number of samples is shown in bold and the expected numbers from a random distribution are in the parentheses. A chi-square test was used to determine possible dependence between the analyzed parameters.

**Figure 3 cells-10-02928-f003:**
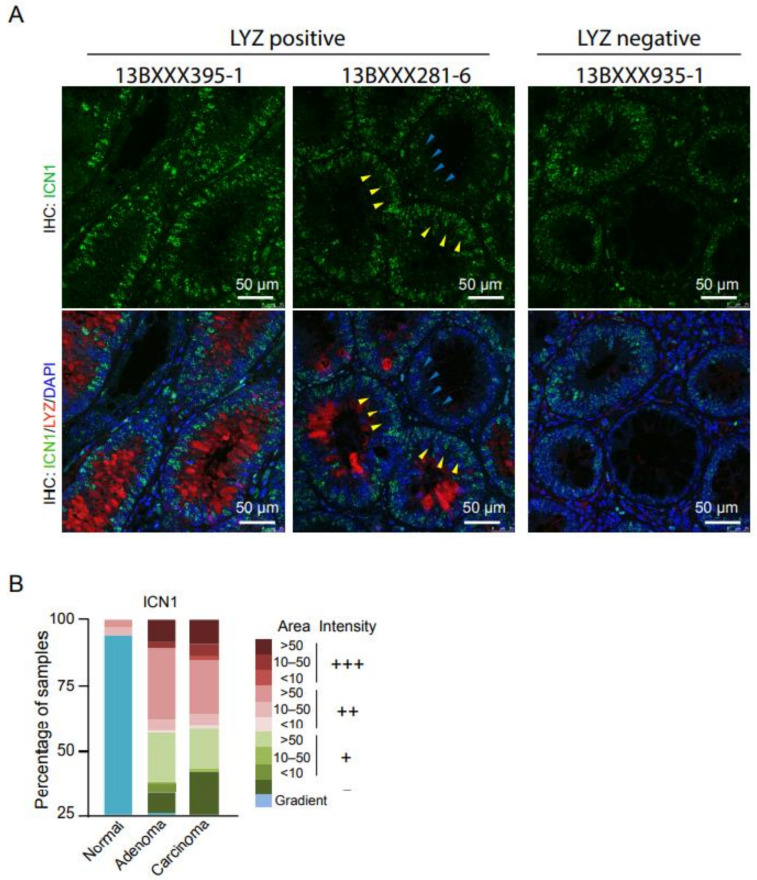
Notch1 activity is higher in adenoma cells of the Paneth-enriched areas. (**A**) Double immunofluorescence (IF) analysis of active Notch1 (ICN1) and LYZ in the indicated samples. DAPI was use for nuclei detection. (**B**) Quantification of the intensity of staining and area of the tissue (normal of tumoral) that was considered positive for active Notch1. Color code indicates the intensity and the area covered by ICN1-positive cells.

**Figure 4 cells-10-02928-f004:**
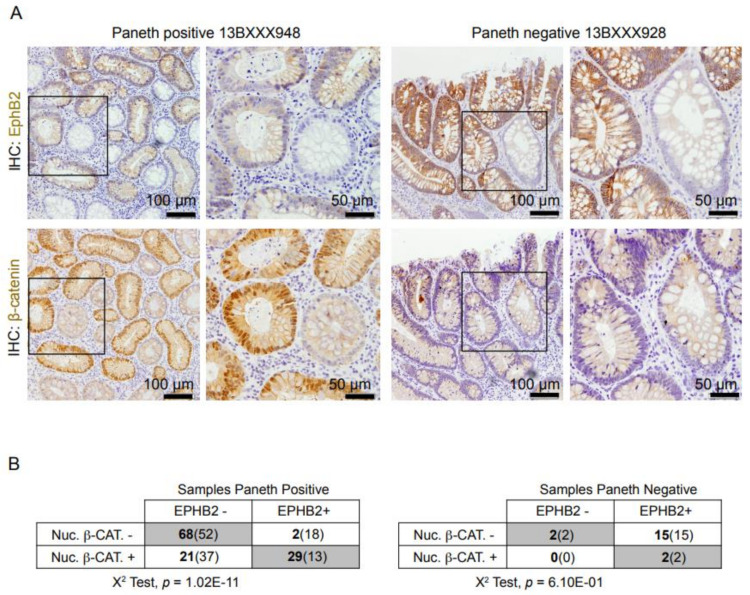
Expression of EPHB2 is linked with Wnt/β-catenin activation in Paneth cell-positive adenomas. (**A**) Immunohistochemical (IHC) analysis of EPHB2 and β-catenin in sequential sections of the indicated adenoma samples. (**B**) Correlation analysis of the presence of nuclear β-catenin and LYZ positivity in the adenoma samples. The actual number of samples for each condition is shown in bold and the expected numbers from a random distribution are in the parentheses. A chi-square test was used to determine possible dependence between the analyzed parameters.

**Figure 5 cells-10-02928-f005:**
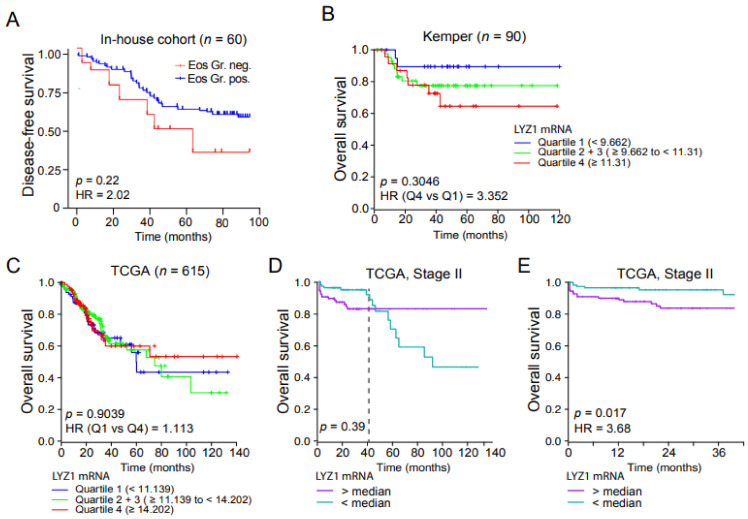
Presence of Paneth cells in the adenomas is indicative of poor prognosis. (**A**) Kaplan–Meier curves representing patients’ disease-free survival classified according to the presence of eosin-positive granules in the adenoma tissue from our in-house cohort of 60 colorectal cancer patients. (**B**,**C**) Kaplan–Meier curves representing patients DFS classified according to LYZ expression levels in patients from the Kemper (**B**) or TCGA (**C**) CRC datasets considering all stages. (**D**,**E**) Kaplan–Meier curves representing patients DFS classified according to LYZ expression levels in stage II patients from TCGA data set at 140 months (**D**) or 40 months (**E**) from diagnosis. We used Cox proportional hazards models for statistical Kaplan–Meier analysis.

## Data Availability

Samples and clinical data from our in-house cohort of patients are deposited at MARBiobank (Hospital de Mar, Barcelona) and can be obtained on demand. Public datasets used in this study are available at the TCGA Portal.

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
