# Peer review of "Accumulation of Paneth Cells in Early Colorectal Adenomas Is Associated with Beta-Catenin Signaling and Poor Patient Prognosis"

_cells, 2021, doi:10.3390/cells10112928_

Round 1

Reviewer 1 Report

The main message of the paper is that the presence of Paneth cells in early colorectal adenomas is associated with increased beta-catenin signaling, which could provide important additional information to clinicians and help them to assess patient’s prognosis and optimal therapy. The data in this paper seems to support this conclusion. However, there are several points where paper should be improved. I see three major points and several minor ones.

Major points:

  • Authors claim that accumulation of Paneth cells in early colorectal adenomas is associated with poor patient prognosis in the title. However, they do not provide significant results to support this claim. Survival analysis in the manuscript does not show any significant difference at p<=0.05. Especially with high number of samples in TCGA dataset this raises the question about validity of this claim. In my opinion the main culprit could be inclusion of all stages from TCGA dataset as the main effect of Paneth cells presence should be at early stages. Overall authors should reanalyze their data and if no significant effect could be shown the association with poor prognosis have to be removed from the title and this fact noted in the text associated with Figure 5.
  • In methods is missing the size of the field where the cells were counted, number of these fields per sample and how position of this field was selected.
  • The statistics used is not clearly described. In methods are mentioned Wilcoxon and Kruskal-Wallis tests, but they are not used in manuscript where only χ2 test is used. χ2 test is not mentioned in methods. The method used for statistical survival analysis must be better described in methods.

Minor points:

  • Abbreviation - “tumor initiating cells” (TICs) – is used only in Introduction and only once and thus unnecessary
  • ISC abbreviation is not defined in the text
  • On line 89 - Primary antibodies (and – also second parenthesis is missing
  • Line 103 - anti-N1IC – but in text is used ICN1
  • Line 157 - LYZ and β-catenin demonstrated a perfect correlation – there is no analysis of correlation (Figure 2D does not show such correlation). Either correlation should be analyzed or the text reworded.
  • Figure 2. Title is the same as Figure 1 title
  • Figure 3B. Color in plot is not in legend

Author Response

Reviewer 1

The main message of the paper is that the presence of Paneth cells in early colorectal adenomas is associated with increased beta-catenin signaling, which could provide important additional information to clinicians and help them to assess patient’s prognosis and optimal therapy. The data in this paper seems to support this conclusion. However, there are several points where paper should be improved. I see three major points and several minor ones.

Major points:

Reviewer: Authors claim that accumulation of Paneth cells in early colorectal adenomas is associated with poor patient prognosis in the title. However, they do not provide significant results to support this claim. Survival analysis in the manuscript does not show any significant difference at p<=0.05. Especially with high number of samples in TCGA dataset this raises the question about validity of this claim. In my opinion the main culprit could be inclusion of all stages from TCGA dataset as the main effect of Paneth cells presence should be at early stages. Overall authors should reanalyze their data and if no significant effect could be shown the association with poor prognosis have to be removed from the title and this fact noted in the text associated with Figure 5.

Answer: We have now analyzed additional datasets and found a trend towards poor prognosis in the Kemper dataset that was generated from stage II CRC samples. Similar, but significant, results were obtained by the analysis of stage II TCGA patients in the initial 40 moths after diagnosis. These results are now included in new Figures 5B-E and in the text: “Similarly, patient stratification based on LYZ mRNA levels using the stage II Kemper CRC dataset34 denoted a poor prognosis value of this marker (p=0.30; HR=3.35) (Figure 5B) that was not observed in the TCGA CRC dataset (Figure 5C). Interestingly, restricting the analysis to stage II patients in the TCGA dataset patients, LYZ-high tumors show a robust trend towards poor prognosis in the first 40 months after diagnosis (Figure 5D). Analysis of overall survival in this initial period confirmed the prognosis value of the LYZ marker (Figure 5E).”

Reviewer: In methods is missing the size of the field where the cells were counted, number of these fields per sample and how position of this field was selected.

Answer: We have now included this information in lane 115: “In all the analyses a minimum of 5 fields containing tumour areas were randomly selected and evaluated by two independent researchers. Images at 400x magnification were used to determine the percentage of tumor cells that exhibited positive immunoreactivity for the different antibodies. In Tissue Microarray samples, staining intensity was assessed directly under the microscope by two independent researchers, at 100x magnification, but amplifying tumour areas at 200x when necessary to obtain agreement among researchers.”

Reviewer: The statistics used is not clearly described. In methods are mentioned Wilcoxon and Kruskal-Wallis tests, but they are not used in manuscript where only χtest is used. χtest is not mentioned in methods. The method used for statistical survival analysis must be better described in methods.

Answer: We have included this information in the methods section in lane 126: “Pearson’s Chi-Square test with a cross tabulation table of 2x2 was performed to determine the statistical association between marker expression (positive or negative) using Excel. R score for correlation (Pearson coefficient) was used to test the correlation between quantitative measurements of marker expression (0-1-2-3 or beta-catenin and 0-1 for Lysozyme) using Graphpad Prism 9; p value was calculated using a two-tailed confidence interval of 95%. Survival analyses were performed on Graphpad Prism 9, to measure 5-year disease free survival; X values were measured in months, Y values were coded 1 for death/event, 0 for censored subjects; survival curves were compared using the logrank (Mantel-Cox test) and the Gehan-Breslow-Wilcoxon test.”

Minor points:

-Abbreviation - “tumor initiating cells” (TICs) – is used only in Introduction and only once and thus unnecessary

-ISC abbreviation is not defined in the text

-On line 89 - Primary antibodies (and – also second parenthesis is missing

-Line 103 - anti-N1IC – but in text is used ICN1

-Line 157 - LYZ and β-catenin demonstrated a perfect correlation – there is no analysis of correlation (Figure 2D does not show such correlation). Either correlation should be analyzed or the text reworded.

-Figure 2. Title is the same as Figure 1 title

-Figure 3B. Color in plot is not in legend

Answer: We have revised all this issues.

Reviewer 2 Report

Congratulations on a insightful original article in the role of Paneth cells in early colorectal adenomas and subsequent patient prognosis.

Author Response

We thank the reviewer for the positive comments

Reviewer 3 Report

The paper is very interesting and it offers to Gastrointestinal Pathologists new opportunities to find early signs of a possible neoplastic evolution of the colonic lesions. I consider it fit for the publication 

My anntotations are the following:

1) Page 2 Line 73 Matherials and Methods - a numebre of only 30 patients could be considered not sufficient to perform conclusions
2)  Page 3 line 28 Results- the role of Eosinophilic granuiles should be extended and explained better

Author Response

Reviewer 3

The paper is very interesting and it offers to Gastrointestinal Pathologists new opportunities to find early signs of a possible neoplastic evolution of the colonic lesions. I consider it fit for the publication 

My annotations are the following:

Reviewer: Page 2 Line 73 Matherials and Methods - a number of only 30 patients could be considered not sufficient to perform conclusions.

Answer: We apologize for erroneously mentioning 30 samples, when the initial analysis included 38 samples, which is now corrected. Since we also considered this number of samples insufficient, we perform a second analysis with 200 hundred tumor samples and two independent public datasets.

Reviewer: Page 3 line 28 Results- the role of Eosinophilic granuiles should be extended and explained better.

Answer: We have extended this particular concept. In lane 143: “We have here selected a small discovery cohort of 38 early adenomas characterized by the presence of large eosinophilic granules in the cytoplasm, indicative of Paneth cells.”

Round 2

Reviewer 1 Report

Role of Paneth cells in adenoma progression seems to be interesting topic. For the future analysis it might be interesting to combine information presence of Paneth cells in adenoma with localization of the adenoma.